# Comparative Assessment of the Shaping Ability of Reciproc Blue, WaveOne Gold, and ProTaper Gold in Simulated Root Canals

**DOI:** 10.3390/ma15093028

**Published:** 2022-04-21

**Authors:** Laura Orel, Oana-Alexandra Velea-Barta, Cosmin Sinescu, Virgil-Florin Duma, Luminița-Maria Nica, Razvan Mihai Horhat, Raul Dorin Chirila, Anca Tudor, Dan-Dumitru Vulcănescu, Meda Lavinia Negrutiu

**Affiliations:** 13rd Department, Discipline of Odontotherapy and Endodontics, Faculty of Dental Medicine, Research Center in Dental Medicine Using Conventional and Alternative Technologies, “Victor Babes” University of Medicine and Pharmacy, 300041 Timisoara, Romania; orel.laura@umft.ro; 23rd Department, Discipline of Odontotherapy and Endodontics, Faculty of Dental Medicine, “Victor Babes” University of Medicine and Pharmacy, 300041 Timisoara, Romania; 31st Department, Discipline of Prosthesis Technology and Dental Materials, Faculty of Dental Medicine, Research Center in Dental Medicine Using Conventional and Alternative Technologies, “Victor Babes” University of Medicine and Pharmacy, 300041 Timisoara, Romania; minosinescu@yahoo.com (C.S.); medanegrutiu@gmail.com (M.L.N.); 43OM Optomechatronics Group, Faculty of Engineering, “Aurel Vlaicu” University of Arad, 310130 Arad, Romania; duma.virgil@osamember.org; 5Doctoral School, Polytechnic University of Timisoara, 1 Mihai Viteazu Ave., 300222 Timisoara, Romania; 63rd Department, Discipline of Odontotherapy and Endodontics, Faculty of Dental Medicine, TADERP Research Center, “Victor Babes” University of Medicine and Pharmacy, 300041 Timisoara, Romania; razvanhorhat@yahoo.com; 7Faculty of General Medicine, “Victor Babes” University of Medicine and Pharmacy, 300041 Timisoara, Romania; dorin.raul@gmail.com; 8Discipline of Computer Science and Medical Biostatistics, “Victor Babes” University of Medicine and Pharmacy, 300041 Timisoara, Romania; atudor@umft.ro; 9Emergency Hospital for Children “Louis Țurcanu”, Str. Dr. Iosif Nemoianu, No. 2, 300011 Timisoara, Romania; dannvulcanescu@gmail.com

**Keywords:** root canal shaping, simulated root canals, centering ability, standardized photographs, blue wire NiTi, gold wire NiTi, heat treatment

## Abstract

Maintaining the original trajectory of the root canal is a major challenge in endodontic therapy, especially in narrow and curved root canals. The present study aims to assess the shaping capacity of three endodontic systems made of different nickel–titanium alloys on simulated curved root canals. Thirty-six endodontic resin blocks (Ref. V040245, VDW) divided into three groups, each of twelve blocks (*n* = 12), were shaped, photographed, and analyzed: Group 1-Protaper Gold (PTG) (Dentsply Maillefer, Ballaigues, Switzerland) F2 25/08; Group 2-Reciproc Blue (RB), RB 25/08 (VDW, Munich, Germany); Group 3-WaveOne Gold (WOG) (Dentsply Maillefer), WOG 25/07. Each block was standardized and photographed before and after shaping in the same position, with the foramen oriented to the left. Post-shaping images were superimposed onto the initial ones. Thirteen measurement points were used for evaluation, spaced with 1 mm distance from one another, from level 0, apical foramen, to level 12, coronal orifice. The amount of removed resin from inner (X1) and outer (X2) walls, the direction of transportation (X1 − X2), and the centering ability (X1 − X2)/Y were measured, calculated, and comparatively analyzed. Statistical differences (*p* < 0.05) were observed between the shaping capacity of the considered systems in the middle and coronal thirds. PTG had a better centering ability than WOG and RB in the coronal third, while RB was more centered in the middle third in comparison to both WOG and PTG. In the apical third, the centering capacity of WOG was higher, without being statistically significant. WOG 25/07 and PTG 25/08 tend to cut more on the inner wall of the root canals, and RB 25/08 on the external one.

## 1. Introduction

One of the most significant steps of the root canal treatment is represented by the preparation of the root canal system [1]. The instrumentation of the root canal is a key stage in the endodontic treatment, and it is considered a predictive factor for the long-term success of endodontic therapy when correctly accomplished. Ideally, the mechanical preparation of the endodontic space should give to the root canal a continuous tapered shape from the coronal level to the apical third, respecting the anatomical shape and multiplanar curves of the canal and keeping the size of the foramen as small as practical [2].

Achieving a continuous tapered shape and respecting the original morphology in terms of narrow and curved root canals is still a major challenge in current practice. Traditionally, mechanical canal instrumentation using stainless steel hand tools is a time-consuming method, with which it is difficult to meet the above criteria when preparing narrow and curved canals [3].

Recently, several modern endodontic nickel–titanium (NiTi) systems with different mechanical characteristics have been designed to improve the behavior of endodontic instruments [4]. The continuous research in this field and the deepening of knowledge related to the NiTi alloys has led to the development of different types of instruments with superior shaping characteristics. As a result, more appropriate preparations in accordance with the natural shape of the root canal have been achieved. Consequently, the incidence of iatrogenic errors such as canal transportation, zipping, perforations, and ledges has been seriously reduced [1,5].

Thus, to increase the efficiency and safety of the mechanical preparation of the root canal, the companies that manufacture NiTi instruments have been constantly focused on improving their physical properties by increasing their flexibility and the resistance threshold to bending and fracture. These new approaches and changes include not only different tapers and designs of the cross-section of the instruments, but also novel NiTi alloys to improve their mechanical characteristics [6,7].

ProTaper Gold (PTG), WaveOne Gold (WOG), and Reciproc Blue (RB) are relatively novel root canal shaping systems. They were developed from the improvement of NiTi alloys [8]. Literature states that the heat treatment of the WOG NiTi alloy represents a slow heating–cooling process, which generates Ti3Ni4 precipitates dispersed over its surface [9]. This offers the instruments a higher resistance to cyclic fatigue in comparison to M-wire reciprocating files [10,11]. In contrast, the blue treatment of RB involves a heating–cooling process that generates a blue colored titanium oxide layer [12,13]. These heat-treated RB instruments provide greater flexibility and resistance to cyclic fatigue when compared to the classical Reciproc (VDW GmbH) files made from M-wire [14,15,16].

Different studies have been already carried out to investigate such aspects, either on simulated root canals [2,3,4,5], in vitro on extracted human teeth [16,17], or in silico, for which molar models are digitized for 3D simulations. The Finite Element Analysis (FEA) is a mathematical approach properly utilized in dentistry and represents a useful tool to simulate the mechanical behavior of dental materials [18]. Moreover, it has been recently used to describe the role of the elasticity of the alloy in the shaping behavior of NiTi instruments [19].

The aim of the present study was to assess the shaping efficiency of two reciprocating NiTi systems, RB and WOG, as well as of one system with continuous rotation, PTG, on endodontic resin blocks with simulated curved root canals. The standardized photographic method was chosen to obtain before/after shaping images, which were digitally processed and analyzed using a computerized program for precise measurements. Statistical analysis was performed to observe the differences between the study groups.

The tested null hypothesis was that no differences can be observed between the shaping capacity of the three tested systems.

## 2. Materials and Methods

The present study utilized thirty-six endodontic resin blocks, (Ref. V040245, VDW, Munich, Germany) with simulated curved root canals. They were divided into three groups, each consisting of twelve resin blocks (*n* = 12). The samples from Group 1 were shaped with the Protaper Gold system (Dentsply Maillefer, Ballaigues, Switzerland), instruments S1-S2-F1-F2, up to a final size of 25/08; the blocks from Group 2 were shaped with the Reciproc Blue system (VDW, Munich, Germany) instrument RB 25/08; in Group 3 the WaveOne Gold system (Dentsply Maillefer, Ballaigues, Switzerland), instrument WOG Primary 25/07 was utilized.

The working length of each root canal was determined by using an ISO 10 K-file and an endodontic ruler, and a reproducible glide path was achieved using 10 hand K-files in push-pull/filling motions until the files were loose in the root canal, before using any rotary/reciprocating instrument. Each resin block was photographed before and after shaping. The images were taken on a fixed stand with a Nikon D7000 camera (Nikon, Tokyo, Japan) and an AF-S MICRO NIKKON 60 MM 1:2:8G ED camera lens in a standardized manner. The improvised stand was used to ensure that all the resin blocks were fixed in the exact same position, with the foramen to the left with regard to the sensor of the camera. All pictures were taken under the same conditions. Each root canal was injected with Castellani red dye solution to make the root canal trajectory more visible on the before/after photographs.

An X-Smart Plus (Dentsply Maillefer, Ballaigues, Switzerland) endodontic motor was used for the shaping phase, with the corresponding settings of the manufacturer for each file. During shaping, the root canals were filled with NaOCl 5.25% irrigant solution 1:1 (Chloraxid, Cerkamed, Stalowa Wola, Poland).

Group 1 (*n* = 12) was shaped with the PTG rotary system, using the S1-S2-F1-F2 sequence; therefore, the final shape had the size of 0.25 mm at the foramen and 8% apical taper. Between each instrument of the PTG system, the canals were irrigated with NaOCl. The patency was re-confirmed with K-file #10, and then re-irrigated with the same solution; therefore, during shaping, canals were permanently filled with irrigant.

Group 2 (*n* = 12) was shaped using the RB system, using the RB25 instrument with a 0.25 tip diameter and 8% taper on the first 3 mm. Three passages of the RB25 instrument were used until the complete shaping of each root canal in reciprocating motion was completed, according to the manufacturers’ instructions. Irrigation and patency were recapitulated before each passage of the RB25 instrument inside the root canal.

Group 3 (*n* = 12) was shaped using the WOG endodontic system, instrument WaveOne Gold Primary, with a tip diameter of 0.25 mm and a 7% taper, in the same manner as for Group 2, until the WOG 25/07 instrument reached the working length.

All samples were shaped by only one experienced Endodontics specialist (L.O.) to eliminate any variables regarding the shaping ability of different operators.

All photographs taken after shaping were processed using the Adobe Photoshop CC (Adobe Systems Inc., San Jose, CA, USA) program, by overlaying (i.e., superimposing) the final images over the initial ones. As a result, images with the differences between the initial morphology of the root canal and the final shape were obtained. The obtained images were cropped to certain dimensions in order to enclose them into a standardized frame of 1054 × 442 px (15.5 × 6 mm). Each image was then measured with the Adobe Photoshop Measuring Tool. This allowed for the exact measurement of the investigated distances in mm, which represents an advantage of the current study. The measurements were made at thirteen different levels, with a 1 mm distance between each level (Figure 1).

The images were then structured depending on the utilized file and the obtained result with Microsoft Excel program, in documents that contained all the measurements, in a non-blinded study. The distance between the initial root canal margin and the margin after shaping on the internal side of the curvature (X1), the external side of the curvature (X2), as well as the total width of the shaped root canal (Y) were measured. Using the following equations, the obtained data served for a qualitative and quantitative assessment of the modelling capacity of the three considered systems by evaluating the following parameters:the amount of resin removed on the inner part of the curvature, X1the amount of resin removed on the outer part of the curvature, X2the total amount of resin removed, X1 + X2the amount and direction of transportation, X1 − X2the centering ratio (X1 − X2)/Y, where Y is the total width of the shaped root canal.

The closer the centering ratio was to zero, the better the centering ability of a certain system was considered [2].

No instrument fractured during shaping. All root canals remained patent by using the 0.10 ISO K-file in recapitulation after each used NiTi instrument associated with co-pious irrigation.

## 3. Results

The obtained results are based on the total number of thirty-six endodontic resin blocks included in the study. In each group, face L (left) of the plastic blocks was photographed and evaluated.

Measurements were made on all photographs taken for the three-dimensional (3D) reproduction of changes resulting after instrumentation from mm to mm along the entire root canal length. Thus, thirteen levels of analysis were obtained, from 0 to 12. On each level, the parameters pointed out above were measured. The results were comparatively analyzed for each root canal third, as follows: From level 0 to 4, the apical third; from 5 to 8, the middle third; from 9 to 12, the coronal third (Figure 2).

All data was recorded using Microsoft Excel. Statistical analysis of the data was performed using SPSS 22.0 (SPSS Inc., Chicago, IL, USA). Each set of measurements was analyzed using the Kolmogorov–Smirnov test for normality.

Statistical significance level was set at *p* < 0.05. Descriptive statistics (mean and standard deviation) were provided for all data sets. For a comparison between data sets, the ANOVA test was performed. Tukey’s post hoc test was also performed to determine the exact location of the differences between groups.

### 3.1. Statistical Analysis of the Apical Third (Levels 0 to 4)

In the apical third of the simulated root canals (levels 0 to 4), no statistically significant differences were recorded between the three analyzed systems for X1, X2, and for (X1 − X2)/Y. However, as it can be observed (Table 1), RB removed the highest amount of resin from the inner part of the curvature, in the first millimeters of the root canal, starting from the foramen level (level 0) to level 2, in comparison to both WOG and PTG. At levels 3 and 4, WOG removed more resin than RB and PTG. On the inner part of the curvature in the apical region, PTG was the most conservative system, with the lowest amount of resin removed at all five analyzed points.

On the external part of the curvature, RB removed the highest amount of resin in comparison to WOG and PTG at all five analyzed levels, while WOG was the most conservative system at levels 3 and 4, and PTG at levels 0 to 2, but the differences observed between all three systems were not statistically significant.

### 3.2. Statistical Analysis of the Middle Third (Levels 5 to 8)

In the middle third of the root canals, the most statistically significant differences were observed between the three systems in comparison to the apical and coronal thirds. For X1 (Table 2), significant differences were observed at levels 6, 7, and 8. At level 6, this is due to the differences between the RB and WOG groups, the mean value of X1 for RB being the lowest. At levels 7 and 8, the RB mean X1 value was lower in comparison to both WOG and PTG. These results show that RB removed less resin from the inner part of the root canal in the middle third when compared to the other two systems.

For X2 (Table 2), differences in the ANOVA test were recorded at all analyzed levels, as the RB mean was higher than both those of WOG and PTG. This means that RB removed the highest amount of resin from the external part of the root canal in the middle third in comparison to both WOG and PTG.

For the (X1 − X2)/Y formula, statistically significant differences were recorded at all levels. At level 5, the mean value for RB was lower only compared to WOG. At level 6, RB had a value closer to 0, while at level 7 and 8, PTG and WOG had values closer to 0. These values mean that they had a more centered preparation at the specified level.

### 3.3. Statistical Analysis of the Coronal Third (Levels 9 to 12)

In the coronal part of the root canals, several statistically significant differences were also observed. For X1 (Table 3), differences were recorded at levels 10 and 12. At level 10, the mean for RB was lower than for WOG, while at level 12 it was lower than both for WOG and PTG. For X2, only at level 9 was the mean for RB lower than for PTG. For the (X1 − X2)/Y formula, there were differences at levels 9 and 10, with the mean for RB being lower than for PTG (Table 3).

Moreover, an overall comparative analysis of the efficiency of the considered systems on each root canal third was performed (Table 4). Regarding the centering capacity, in the apical third WOG and PTG showed values closer to 0, but no statistically significant differences were recorded in comparison to RB. In the middle third, the value for RB was the lowest, while no differences were observed between WOG and PTG. In the coronal third, WOG and PTG had values closer to 0 and therefore the most centered preparations.

## 4. Discussion

According to extensive literature data, despite the limitations of the present study, it is widely known that image superposition is an accepted method utilized to assess the shaping efficiency of different instrumentation methods on resin blocks with simulated root canals [20,21,22]. However, several other methods can be utilized to evaluate and compare root canal morphology before and after instrumentation either on extracted teeth or tooth replicas. They include radiography, cone beam computed tomography, micro-CT, 3D-FEA, etc. [19,23,24].

Resin blocks were chosen in the present study instead of extracted teeth to rule out variations in the anatomy of the root canal that could influence the result of the instrumentation. The resin blocks ensure the standardization of the diameter, length, and curvature of the root canal in all three dimensions. They also facilitate the modeling capacity of various instruments to be directly compared. There are two significant limitations associated with the use of resin blocks: First, it is important to take care to extrapolate the effects to the clinical situation; second, the heat produced in the resin blocks by the rotating instruments may soften the resin material and lead to the binding of the cutting blade and to the instrument being separated, as the hardness of the resin blocks is half of that of natural human dentine and exhibits completely different thermal properties [8,25]. None of these incidents happened during shaping of the plastic resin blocks, as continuous irrigation was used to prevent over-heating. Thus, a comparison of the resulting shape obtained by using artificial root canals from resin blocks could be a reliable replacement method for natural teeth if the conditions are the same for all tested instruments or techniques [26,27,28].

In most of the investigations performed on radiographs or superimposing photographic before/after images of the plastic blocks, as in the present study, the investigators used mathematical formulas to assess the differences between pre-operative and post-operative outer lines. However, this method of examination does not specify the exact amount of material removed, or the effects of the preparation symmetry, since these investigations are only two-dimensional (mesio-distal), and there is also another dimension (bucco-lingual) in the samples [29]. Due to the existence of the curvature in the apical third, the photographs chosen for the present study were those taken from the left side of the resin block.

This study aimed to compare three different mechanical systems made of various NiTi alloys, such as the blue wire and the gold wire NiTi, which are characterized by an advanced metallurgy and heat treatment process that increases the instruments flexibility. In extended research, the effects of the thermo-mechanical treatments applied to NiTi alloys have been evaluated using differential scanning calorimetry [30,31].

Previous studies compared the modelling capacity of gold wire, M-wire, and traditional NiTi alloys. They indicated that gold wire and M-wire alloys showed substantially less canal transport, which was attributed to the higher flexibility of the alloys [32]. Several studies have shown that more flexible instruments generate more centered root canal preparation. The ProTaper Gold system has a more oriented type of instrumentation in the curved portion compared to its predecessor, the Universal ProTaper system [5,22]. Reciproc Blue receives an advanced heat treatment that changes the molecular structure of the NiTi alloy and offers the instrument a blue color and greater flexibility [32]. Literature data indicates that this thermal treatment technology has also improved the durability of the instruments [14,33].

Temperature is considered to be one of the crucial factors that influences the mechanical properties of NiTi alloy. It also results in differences in their bending properties [34,35]. Researchers and manufacturers are constantly testing the thermo-mechanical procedures used in WOG and RB, aiming to improve the characteristics of the NiTi files [6,36]. Moreover, it is considered that the torsional stiffness of these endodontic instruments is associated with their cutting efficiency [37].

For a better standardization of the shaped samples, all three instruments compared in the present study had a tip diameter equivalent to a size of 0.25 mm and similar taper in the first mm of the active part (0.08 for RB and PTG F2, and 0.07 for WOG Primary). These similar diameters and tapers can explain the similar results obtained in the apical thirds of the root canals, with no significant statistical differences between the shaping capacity of the three compared systems. However, there is a better centering ability of the instruments made of gold alloy, WOG and PTG, in comparison to the blue alloy of RB. As WOG and RB are both reciprocating systems, and PTG is a rotational one, no conclusion can be drawn regarding the superiority of the reciprocation or continuous rotation movement in the apical third.

However, the compared instruments have multiple different tapers along their length in accordance with literature data, which explains their cutting characteristics and the final shapes of the root canal [38,39]. Both RB and WOG have a constant taper on the first 3 mm from the tip (8% for RB and 7% for WOG), and a decreasing taper in rest, to 4% for RB and 3% for WOG Primary at the end of the active part. They are single-file reciprocating systems; therefore, the final shape of the root canal can be achieved by using only one of these instruments [8,9,32].

In comparison, the PTG system uses multiple files with various sizes and tapers to cut the final shape of the root canal in continuous rotation movement in a sequence where all instruments need to reach the working length one before another (S1-S2-F1-F2). The canal is first progressively enlarged with shaping instruments with a fixed small taper at the tip (2% for S1 and 4% for S2), but with a progressive increased taper at the end of the active part (up to 11%), which mainly enlarges the coronal and middle thirds of the root canal [4,8]. Then, F1 and F2 finishing instruments (with a constant bigger taper at the tip (7% and 8% respectively) and decreasing taper in rest) are cutting the final shape, enlarging only the apical third [4,8]. While using a multiple file sequence, no differences were observed between the shaping capacity of the PTG system and the single file system WOG. The only differences recorded in the present study were in the middle and coronal thirds of the root canals between RB and PTG, and between RB and WOG, respectively.

The shaping capacity of four single file systems with different tapers was compared in a recent study that revealed that more tapered instruments remove more resin compared to less tapered instruments, and that the taper of the instruments is the predetermining factor for the shaping capacity of the tested instruments [40]. Moreover, the literature shares studies that compared apical transport with rotational and reciprocal movements in rotary instruments. The supremacy of reciprocal motion was stated by most studies [41,42,43,44]. Wu et al. found no differences between the two movements in apical transport [45], while other studies have shown that rotational motion contributes to less apical transport [46,47]. In accordance with most studies, the reciprocal motion seems to be more effective in preventing apical transport. Due to the following reasons, single-file rotary systems have received an increased attention: Fewer procedural errors, less planning time, and simplified implementation [48,49]. As no significant statistical differences were observed between the three compared systems in the apical third, the present study cannot conclude that reciprocal or continuous rotation motion induces less apical transportation.

One of the most important factors that contributes to a good shaping of the root canal is the ability of the instrument to remain in the center of the root canal space without causing iatrogenic errors [8]. The present study revealed no significant differences between the three studied shaping systems in the apical third of the simulated root canals (levels 0 to 4).

The most statistically significant differences were observed in the middle third of the root canals in comparison to both apical and coronal thirds. RB removed a lower amount of resin from the inner wall of the simulated root canals, when compared to WOG and PTG. This finding is in accordance with literature data which show that more flexible instruments tend to prepare the root canal in a more centered way [5,22]. While the external wall was enlarged more using the RB compared to both WOG and PTG, RB showed the most centered preparation in this third.

In the coronal third, PTG and WOG had a more centered preparation, which is a similar finding to other studies, which found coronal and middle thirds to be better prepared by WOG [50]. RB removed a smaller amount of resin from the inner wall in the coronal and medium thirds compared to WOG and PTG, and a larger amount of resin from the external one.

As results of the present study, the null hypothesis formulated in the beginning was rejected for the middle and coronal thirds of the root canals, where differences were observed between the shaping characteristics of the three compared systems and confirmed for the apical third of the root canal, where PTG, RB, and WOG acted in a similar manner.

## 5. Conclusions

Within the limitations of the present study, it was observed that, regarding the amount of resin removed from the interior and exterior walls of the resin blocks, WOG and PTG tend to cut more on the inner wall of the root canals, while RB acts more on the outer wall. Significant differences were recorded in the coronal and medium thirds when comparing RB to PTG and WOG, while in the apical third, all three considered systems tend to prepare in the same manner. No differences were observed between the shaping capacity of WOG and PTG. Neither of the studied instruments had the same centering ability along the entire length of the simulated root canal.

Future work on this topic includes investigations of such NiTi instruments [51] and of their effects, with ex vivo, but also with in vivo imaging methods, the latter with Optical Coherence Tomography (OCT) as well to allow for non-invasive assessments of dental treatments on patients [52].

## Figures and Tables

**Figure 1 materials-15-03028-f001:**
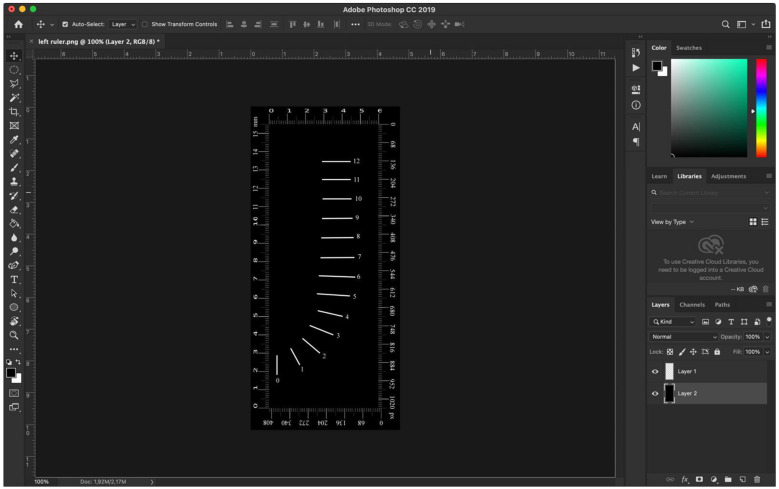
Measuring ruler for the thirteen evaluated levels in mm and pixels (Adobe Photoshop CC-License Type: Subscription, Serial number: 96040415901643637295).

**Figure 2 materials-15-03028-f002:**
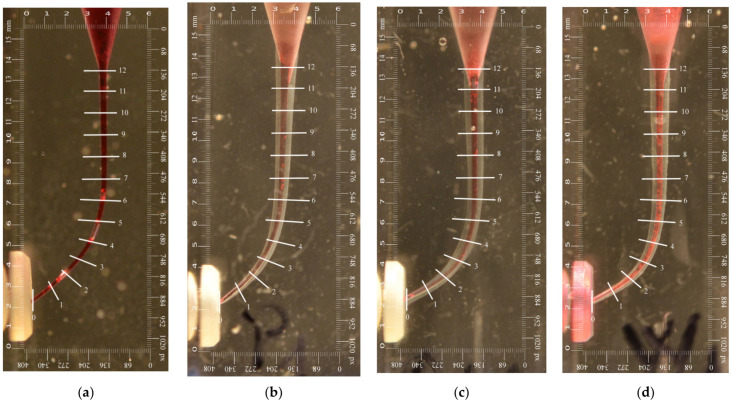
Images of the simulated root canals before/after shaping (examples). (**a**) Initial contour of one of the root canals before shaping. Root canal after shaping with (**b**) PTG (Group 1); (**c**) RB (Group 2); (**d**) WOG (Group 3).

**Table 1 materials-15-03028-t001:** X1, X2, and X1 − X2/Y mean value, standard deviation SD and ANOVA with Tukey’s post hoc test for the apical third.

X1			0	1	2	3	4
	PTG	Mean ± SD	0.055 ± 0.024	0.060 ± 0.013	0.095 ± 0.012	0.128 ± 0.034	0.196 ± 0.042
	RB	Mean ± SD	0.080 ± 0.098	0.100 ± 0.116	0.124 ± 0.139	0.188 ± 0.169	0.200 ± 0.138
	WOG	Mean ± SD	0.061 ± 0.044	0.067 ± 0.030	0.119 ± 0.048	0.193 ± 0.075	0.264 ± 0.061
	ANOVA		0.6051	0.3337	0.6731	0.2789	0.1310
	Tukey post hoc	RB vs. WOG	0.7260	0.4797	0.9000	0.9000	0.2051
		RB vs. PTG	0.5976	0.3502	0.6708	0.3787	0.9000
		WOG vs. PTG	0.9000	0.9000	0.7526	0.3213	0.1679
**X2**							
	PTG	Mean ± SD	0.073 ± 0.030	0.101 ± 0.030	0.127 ± 0.023	0.153 ± 0.028	0.140 ± 0.038
	RB	Mean ± SD	0.0123 ± 0.094	0.136 ± 0.112	0.167 ± 0.159	0.165 ± 0.161	0.178 ± 0.138
	WOG	Mean ± SD	0.108 ± 0.036	0.134 ± 0.032	0.147 ± 0.041	0.111 ± 0.037	0.123 ± 0.097
	ANOVA		0.1217	0.3881	0.5963	0.3686	0.4059
	Tukey post hoc	RB vs. WOG	0.8009	0.9000	0.8557	0.3680	0.3895
		RB vs. PTG	0.1134	0.4420	0.5637	0.9465	0.6316
		WOG vs. PTG	0.3262	0.4758	0.8557	0.5490	0.9122
**(X1 − X2)/Y**							
	PTG	Mean ± SD	−0.053 ± 0.062	−0.111 ± 0.080	−0.074 ± 0.065	−0.049 ± 0.107	0.096 ± 0.127
	RB	Mean ± SD	−0.109 ± 0.423	−0.103 ± 0.551	−0.096 ± 0.635	0.039 ± 0.586	0.039 ± 0.453
	WOG	Mean ± SD	−0.116 ± 0.182	−0.167 ± 0.124	−0.064 ± 0.150	0.143 ± 0.189	0.249 ± 0.180
	ANOVA		0.8234	0.8745	0.9781	0.4332	0.2332
	Tukey post hoc	RB vs. WOG	0.9992	0.8792	0.9750	0.7514	0.2259
		RB vs. PTG	0.8595	0.9964	0.9866	0.8337	0.8798
		WOG vs. PTG	0.8399	0.9142	0.9981	0.4054	0.4604

**Table 2 materials-15-03028-t002:** X1, X2, and X1 − X2/Y mean value, standard deviation SD and ANOVA with Tukey’s post hoc test for the middle third.

X1			5	6	7	8
	PTG	Mean ± SD	0.252 ± 0.038	0.258 ± 0.028	0.248 ± 0.026	0.258 ± 0.037
	RB	Mean ± SD	0.213 ± 0.107	0.208 ± 0.077	0.183 ± 0.076	0.198 ± 0.065
	WOG	Mean ± SD	0.285 ± 0.064	0.288 ± 0.074	0.255 ± 0.055	0.266 ± 0.046
	ANOVA		0.0746	0.0136 *	0.0056 *	0.0042 *
	Tukey post hoc	RB vs. WOG	0.0602	0.0107 *	0.0092 *	0.0064 *
		RB vs. PTG	0.4180	0.1412	0.0190 *	0.0178 *
		WOG vs. PTG	0.5271	0.4922	0.9000	0.9000
**X2**						
	PTG	Mean ± SD	0.123 ± 0.039	0.131 ± 0.029	0.179 ± 0.040	0.203 ± 0.042
	RB	Mean ± SD	0.183 ± 0.112	0.231 ± 0.087	0.265 ± 0.093	0.287 ± 0.068
	WOG	Mean ± SD	0.100 ± 0.055	0.138 ± 0053	0.185 ± 0.050	0.230 ± 0.040
	ANOVA		0.0313 *	0.0004 *	0.0066 *	0.0013 *
	Tukey post hoc	RB vs. WOG	0.0020 *	0.0020 *	0.0137 *	0.0292 *
		RB vs. PTG	0.0009 *	0.0009 *	0.0078 *	0.0011 *
		WOG vs. PTG	0.9615	0.9615	0.9739	0.4243
**(X1 − X2)/Y**						
	PTG	Mean ± SD	0.215 ± 0.119	0.201 ± 0.086	0.103 ± 0.094	0.075 ± 0.101
	RB	Mean ± SD	0.050 ± 0.350	−0.032 ± 0.249	−0.115 ± 0.235	−0.117 ± 0.170
	WOG	Mean ± SD	0.303 ± 0.179	0.228 ± 0.175	0.100 ± 0.134	0.041 ± 0.107
	ANOVA		0.0409 *	0.0023 *	0.0032 *	0.0021 *
	Tukey post hoc	RB vs. WOG	0.0331 *	0.0039 *	0.0082 *	0.0134 *
		RB vs. PTG	0.2124	0.0097 *	0.0076 *	0.0027 *
		WOG vs. PTG	0.6354	0.9358	0.9994	0.8142

* Statistically significant differences.

**Table 3 materials-15-03028-t003:** X1, X2, and X1 − X2/Y mean value, standard deviation SD and ANOVA with Tukey’s post hoc test for the coronal third.

X1			9	10	11	12
	PTG	Mean ± SD	0.254 ± 0.041	0.259 ± 0.036	0.248 ± 0.050	0.161 ± 0.074
	RB	Mean ± SD	0.216 ± 0.055	0.221 ± 0.067	0.220 ± 0.051	0.072 ± 0.056
	WOG	Mean ± SD	0.164 ± 0.043	0.291 ± 0.037	0.249 ± 0.084	0.164 ± 0.060
	ANOVA		0.0523	0.0050 *	0.4506	0.0014 *
	Tukey post hoc	RB vs. WOG	0.0620	0.0035 *	0.5080	0.0034 *
		RB vs. PTG	0.1264	0.1454	0.5263	0.0047 *
		WOG vs. PTG	0.9000	0.2615	0.9000	0.9000
**X2**						
	PTG	Mean ± SD	0.236 ± 0.024	0.252 ± 0.030	0.260 ± 0.052	0.158 ± 0.061
	RB	Mean ± SD	0.305 ± 0.072	0.296 ± 0.061	0.290 ± 0.060	0.116 ± 0.064
	WOG	Mean ± SD	0.261 ± 0.045	0.287 ± 0.037	0.271 ± 0.044	0.151 ± 0.054
	ANOVA		0.0077 *	0.0505	0.3736	0.1991
	Tukey post hoc	RB vs. WOG	0.1006	0.8703	0.6450	0.3327
		RB vs. PTG	0.0060 *	0.0534	0.3493	0.2151
		WOG vs. PTG	0.4605	0.1487	0.8680	0.9594
**(** **X1 − X2)/Y**						
	PTG	Mean ± SD	0.022 ± 0.064	0.009 ± 0.051	−0.027 ± 0.081	0.005 ± 0.080
	RB	Mean ± SD	−0.111 ± 0.152	−0.089 ± 0.139	−0.079 ± 0.114	−0.045 ± 0.097
	WOG	Mean ± SD	−0.009 ± 0.099	−0.004 ± 0.076	−0.029 ± 0.077	0.008 ± 0.047
	ANOVA		0.0162 *	0.0366 *	0.3131	0.1790
	Tukey post hoc	RB vs. WOG	0.0752	0.0852	0.4130	0.2269
		RB vs. PTG	0.0177 *	0.0459 *	0.3729	0.2573
		WOG vs. PTG	0.8031	0.9555	0.9969	0.9968

* Statistically significant differences.

**Table 4 materials-15-03028-t004:** Overall comparison of the mean values between PTG, RB, and WOG in the apical, middle, and coronal thirds.

		PTG	RB	WOG	ANOVA		Tukey Post-Hoc	
Apical third						RB vs. WOG	RB vs. PTG	WOG vs. PTG
	**X1**	0.107	0.139	0.141	0.3803	0.9959	0.4753	0.4256
	**X2**	0.119	0.154	0.125	0.3841	0.5306	0.3974	0.9712
	(**X1 − X2**)**/Y**	−0.038	−0.046	0.007	0.8600	0.8664	0.9971	0.8995
Middle third								
	**X1**	0.254	0.200	0.274	0.0126	0.0118 *	0.0792	0.6931
	**X2**	0.159	0.241	0.163	0.0028	0.0089 *	0.0058 *	0.9844
	(**X1 − X2**)**/Y**	0.148	−0.054	0.168	0.0057	0.0095 *	0.0192 *	0.9566
Coronal third								
	**X1**	0.2306	0.1821	0.2413	0.0034	0.0043 *	0.0206 *	0.8113
	**X2**	0.2263	0.2517	0.2423	0.2935	0.8379	0.2726	0.5808
	(**X1 − X2**)**/Y**	0.0021	−0.0809	−0.0084	0.0272	0.0730	0.0353 *	0.9420

* Statistically significant differences.

## Data Availability

Data available on request from the correspondent authors.

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
