# Peer review of "Comparative Assessment of the Shaping Ability of Reciproc Blue, WaveOne Gold, and ProTaper Gold in Simulated Root Canals"

_materials, 2022, doi:10.3390/ma15093028_

Round 1

Reviewer 1 Report

The manuscript looks interesting and original in the research field. But some improvements are necessary in the different sections as described below to render it available for the publication on Materials. To render easier the work to the Authors the following paper has been submitted to a proper review. Abstract is OK. Introduction must be revised at the suggested points. Please let's modify: at line 55: "root canal systems" in place of endodontic systems; at line 56, "it is considered" in place of represents; at line 58, "to" the root canal; at line 67, "mechanical" characteristics; at 67, performance, do you mean "behavior": please change it; at 70, superior shaping: what do you mean exactly: root canals tapering? cutting efficiency? dentine volume respect; at line 71-75, Please let refer some more recent publications  on this focus; at 76, "physical".... properties; at 79, specify: "mechanical"... properties; at line 81, clean ..mechanical; at 82-85: it is a too long description, please separate and rewrite it;. before line 89 it is convenient to introduce and reference other methods already described in dental literature to investigate mechanical aspects. For example:"Different studies were already  carried out to investigate these aspects, by sound teeth and in vitro, (Shaping ability of reciprocating single-file systems in severely curved canals: WaveOne and  Reciproc versus  WaveOne  Gold  and Reciproc blue. Bürklein S, et al. Odontology. 2019 Jan;107(1):96-102. doi: 10.1007/s10266-018-0364-3. Epub 2018 May 18.) or  "in silico", where molar models were digitized for simulations in 3D.  "FEA (Finite element Analysis) is a mathematical approach properly used in dentistry and represents a useful tool  to simulate mechanical behavior of dental materials (Adhesive class I restorations in sound molar teeth incorporating combined resin-composite and glass ionomer materials: CAD-FE modeling and analysis.Ausiello P, et al. .Dent Mater. 2019 Oct;35(10):1514-1522. ). It is also recently been described  (3D finite element analysis of rotary instruments in root canal dentine with different elastic moduli Prati, C.,et al.  Applied Sciences (Switzerland)2021, 11(6), 2547) the role of alloy elasticity employed in endodontic instrument shaping behavior. In the present  study, the standardized photographic method has been employed.". You can specify here or in the Methods ...  what the different your Technique offers.  But before ending the Introduction, you have to focus the null hyphothesis of the present investigation, which will be confirmed or rejected in the Discussion. Please let's modify these points. Materials and Methods: at line 102, please specify also the characteristics of this resin in terms of E modulus ( E= ...GPa); please specify the profits of your technique involved in the test and also the limits, lines 124-132; Statystical Analyses are ok and SD in the limit spread; at line 236 you have to insert the null hyphotesis acceptance or refusing; at line 241, please fill the FEA references in support also to in vitro dental investigations (Penteado MM, et al. 2019Mechanical behavior of conceptual posterior dental crowns with functional elasticity gradient. Am J Dent. 32(4):165168.). Discussion must briefly revised. Conclusions are OK. REFERENCE LIST NEEDS SOME UPDATES AS DESCRIBED UP. It will contribute to render complete the Manuscript. 

Reviewer 2 Report

This manuscript deals with the clinically relevant topic of the maintaining the original morphology of the root canal during endodontic shaping in curved root canals.

There are some issues  in the present manuscript that need to be addressed before publication:

Introduction:

1)What is the novelty of the paper? 

2)the introduction should be revised because of some redundancy of concept expresses

3)lane 60: “keeping the size of the foramen as small as possible “. Please revise the text for errors

materials:

4)line 103-15: it is not clear how the glide path was performed

5)Please specify the blinded/non blinded groups characteristics

Discussion:

I find several limitations in the discussion.

6) The authors did not take into consideration that of the 3 systems used, one (ProTaper Gold) is a multi-file system, the other two are single file systems.

They should then explain whether and how a gradual increase in channel taper can affect the results of this research.

7) The ProTaper Gold system also uses a continuous rotation movement, while the other 2 systems analyzed in the research use a reciprocating movement.

The difference in movement was mentioned by the authors, but it was not explained how this could affect the outcomes of this research.

Finally, it is not clear whether the results are attributable to differences in alloy or movement of the systematics used, or whether they are attributable to a combination of these characteristics.

Conclusion:

The authors need to provide the strengths of this research not only limitations, in order to give the readers the benefits of this work.

Reviewer 3 Report

It is nice to read this interesting paper aims to evaluate the effects of shaping  ability from 3 new endo instruments in ex vivo. However as authors mentioned in discussion, the warm produced during the preparation can have a big influence on your results. Especially when you did not apply rinse procedures (which is unfortunately not seen in your methods part). This is also strongly suggested by the producers. This could be part of reason for the non-significant results. Besides, it is always good to mention who/how many experimenters have applied ex-vivo test. 

Round 2

Reviewer 3 Report

Significant improvement has been done, congratulations!